# Genetic Algorithm-Based Online-Partitioning BranchyNet for Accelerating Edge Inference

**DOI:** 10.3390/s23031500

**Published:** 2023-01-29

**Authors:** Jun Na, Handuo Zhang, Jiaxin Lian, Bin Zhang

**Affiliations:** 1Software College, Northeastern University, Shenyang 110169, China; 2School of Computer Science and Engineering, Northeastern University, Shenyang 110169, China

**Keywords:** BranchyNet, DNN partitioning, genetic algorithm, distributed DNN inferencing

## Abstract

In order to effectively apply BranchyNet, a DNN with multiple early-exit branches, in edge intelligent applications, one way is to divide and distribute the inference task of a BranchyNet into a group of robots, drones, vehicles, and other intelligent edge devices. Unlike most existing works trying to select a particular branch to partition and deploy, this paper proposes a genetic algorithm (GA)-based online partitioning approach that splits the whole BranchyNet with all its branches. For this purpose, it establishes a new calculation approach based on the weighted average for estimating total execution time of a given BranchyNet and a two-layer chromosome GA by distinguishing partitioning and deployment during the evolution in GA. The experiment results show that the proposed algorithm can not only result in shorter execution time and lower device-average energy cost but also needs less time to obtain an optimal deployment plan. Such short running time enables the proposed algorithm to generate an optimal deployment plan online, which dynamically meets the actual requirements in deploying an intelligent application in the edge.

## 1. Introduction

The distinguishing ability to learn higher-level feature representations at successive nonlinear layers makes deep neural networks (DNNs) widely applied in image classification. With the progression of hardware and learning techniques, DDNs become deeper. This dramatically increases the inference latency. However, the time needed to process an image usually depends on its complexity. For example, it is faster to recognize a person standing in front of a plain blue backdrop than amid a crowd. As the images in real-world datasets always have different classification difficulties, various researchers proposed the model early-exit mechanism and corresponding implementations for accelerating DNN inference by exiting the inferencing process earlier when reaching the required inference accuracy [1,2,3,4]. For example, BranchyNet [1] is a programming framework that implements the model early-exit mechanism. A standard DNN can be resized to its BranchyNet version by adding exit branches with early exit points at certain layer locations. Then, simple images can be classified and exit the network through these early-exit points without going through all the layers of the original DNN, which leads to speedups of about two to five times in inference time, as shown in [1].

Although the BranchyNet can effectively reduce inference time according to the classification complexity of different images by satisfying the required inference accuracy, it increases the model size by adding additional branches. For instance, the authors in [1] add a branch consisting of one convolutional layer and one fully-connected layer into the basic LeNet-5 network. As a result, the total number of layers increases to seven, 1.4 times the original model’s. For AlexNet, the authors add two branches with a total of five additional layers. The number of layers gets to 1.625 times that of the original model. Larger models undoubtedly need more storage for storing and running. Moreover, running these additional branches can also cost more energy. In the above Branchy-LeNet (B-LeNet), a complex image will go through all seven layers before achieving its classification result, which runs one convolutional layer and one fully-connected layer additionally. Similarly, a complex image processed by the Branchy-AlexNet (B-AlexNet) needs to run three convolutional layers and two fully connected layers additionally. Evidently, this costs more energy than when processed by the original networks. Therefore, when applying BranchyNet in real applications, it is crucial to design and implement an efficient application strategy.

In order to effectively apply BranchyNet, the framework DDNNs [5] adopt BranchyNet to distribute DNNs across the cloud, edge, and devices. It divides a BranchyNet into three parts at pre-defined early-exit points. Moreover, the authors propose three different aggregation methods to integrate results from various same-level exit points. Edgent [6,7] jointly applies model early-exit and model partition. It first trains a branchy model at the offline configuration stage. Then, it tries to obtain an optimal partition at the online tuning stage to maximize inference accuracy under the given latency requirement. This approach cuts the DNN according to the current bandwidth state and deploys the two sub-models to a model device and the edge server. The authors in [4] propose an approach to utilize the communication channels efficiently when an edge device with a built-in auxiliary network shares workload with an edge server with a remote principal network. They introduce the dynamic network-sizing technology to vary the depth of the auxiliary network to adjust the amount of workloads to be transferred to the principal network while maintaining overall accuracy. These works only cut the BranchyNet into two or three fixed parts to support collaborated inferencing among cloud, edge, and devices. However, dividing a BranchyNet into multiple pieces for distributed collaboration by a group of robots, drones, vehicles, and other intelligent edge devices is still challenging.

Unlike most existing cloud–edge or edge–device collaborations, this work focuses on collaborative inferencing among several intelligent edge devices. Let us take the example of object identification in public security systems or city surveillance systems illustrated in Figure 1. A pre-trained B-AlexNet is divided into five pieces and deployed into two drones, a camera, a smartphone, and a smart car. To perform the object identification task, the leftmost drone capture images as input to the B-AlexNet and runs the first branch. If the first branch can result in a correct identification result, it will return the identification result to the business system. Otherwise, the process goes back to the first convolutional layer, whose result will be sent to the camera. In turn, the camera will run the inference task assigned to it and send the corresponding output to the smartphone, which performs the second exit branch and obtains corresponding identification results. There are more similar application scenarios, such as distributed fall detection [8] and traffic prediction [9].

The above illustration shows that the DNN partitioning problem is different from the general task offloading problem [10,11,12], which is a topic of interest in edge computing and mobile edge computing. The task offloading problem emphasizes offloading all or part of computing tasks from an edge device to the associated edge server. On the contrary, DNN partitioning aims to split a large computation task into many sub-tasks and distribute them to a group of edge devices to use the free resources of such edge devices fully. As a result, an edge device needs to perform its task with the help of a selected edge server in task offloading, while the edge devices cooperate to perform a task without an edge server after DNN partitioning.

In order to accelerate edge inference, it has been emerging to partition a complete neural network into several parts and deploy them to a group of edge devices [13,14]. Due to the fact that intermediate results no longer need to be further sent to the remote cloud center, all inference tasks will be completed in the edge environment. This ensures data privacy and saves transmission costs. However, unlike only dividing a network into two or three parts, partitioning a network into multiple parts is an NP-hard problem. Although some strategies are trying to split a DNN into several parts effectively [15,16,17], most of them try to select a particular branch to split and deploy rather than partitioning the whole BranchyNet with all branches which reduces the applicability and flexibility of the generated deployment plan.

This paper proposes a novel genetic algorithm (GA)-based BranchyNet partitioning approach for accelerating edge inference. In order to ensure the applicability of the resulting deployment plan, the partitioning problem is defined as a constrained optimization problem to generate an optimal deployment plan under a given amount of available resources. Then, a two-layer chromosome GA is designed to solve the established problem more efficiently. Experiments are taken on partitioning B-AlexNet and B-ResNet proposed in [1]. By comparing with methods proposed in [5,18,19], the proposed GA significantly decreases the inferencing latency. The main contributions of this paper are as follows:Firstly, this paper presents the formulation for estimating the total execution time of a complete BranchyNet, including inferencing and transmitting the intermediate results, and converses the BranchyNet partitioning problem into a constrained optimization problem;Secondly, this paper puts forward a two-layer chromosomes GA. It emphasizes the consistency between partitioning and deployment and divides chromosomes into partitioning and deployment chromosomes accordingly;Finally, this paper presents a comprehensive experiment evaluation by comparing the proposed method with four other typical DNN partitioning approaches based on B-AlexNet and B-ResNet in inferencing performance and algorithm efficiency.

The rest of this paper is organized as follows. Section 2 reviews and classifies existing DNN partitioning approaches. Section 3 presents the problem formulation for achieving optimal BranchyNet partitioning. Then, Section 4 describes the proposed algorithm’s details, including the network’s pre-processing and improving basic GA, and Section 5 provides the corresponding experimental results. Finally, Section 6 concludes this work and outlines possible directions for future research.

## 2. Literature Review

According to the different deployment targets of DNN, existing DNN partitioning methods can be divided into the following two categories.

One way is to divide a DNN into two or three parts for deploying to the edge device, edge server, and remote cloud. For example, the authors in [20] propose to cut a CNN model at the end of the convolutional layer. Then, they allocate and perform the convolutional layers at the edge and the rest of the fully-connected layers in the cloud. Instead of selecting a fixed partitioning point according to the network structure, the authors in [18] observe that ideal fine-grained DNN partition points depend on the layer compositions of the DNN, the particular mobile platform used, the wireless network configuration, and the server load. With this in mind, they propose a lightweight scheduler named Neurosurgeon to automatically partition DNN computation between mobile devices and data centers for either the best end-to-end latency or mobile energy consumption by testing every candidate point after each layer. Refs. [6,21,22] also adopt similar strategies in order to identify the best partition points. However, unlike other works using exhaustive searching, ref. [22] solves the problem by mixed integer linear programming.

The other way is to divide a DNN into more than two parts and deploy them onto multiple devices. There are usually three different perspectives to partition a DNN into multiple parts. Firstly, considering the requirements for storing large inputs or weights, inputs partitioning [23,24] and weights partitioning [25] approaches are proposed. These two kinds of strategies attempt to slice a DNN horizontally, which enables a single-edge device to store part of the input data or weight when it does not have enough storage. Secondly, layer-based partitioning is proposed to solve the depth problem in DNN inference [15,16]. Thirdly, more works are emerging to provide a hybrid solution [14,17,26,27,28,29]. For example, ref. [26] adopts both input partitioning and layer-based partitioning to obtain a group of small enough sub-models. Ref. [17] proposes a grid fashion to fuse and partition layers vertically. Ref. [28] models a DNN as a data-flow graph and transforms the DNN partitioning problem into a graph-partitioning problem.

In summary, there is a common assumption in nearly all of the above works: the DNN is a well-organized linear structure. Although [6,21] added early-exit branches to the original DNN before partitioning, they split the pre-trained BranchyNet into several independent linear DNNs before partitioning. They selected the best partition points by testing all positions after each layer in all these linear DNNs, which, in essence, only selects and deploys part of the original BranchyNet. It is hard to ensure the actual inference accuracy. As far as we know, there are currently few solutions specifically for BranchyNet partitioning. Moreover, more studies have begun to focus on the joint optimization of DNN partitioning and resource allocation [30,31,32]. However, it is still an open and critical challenge.

## 3. Problem Formulation

### 3.1. System Model

As illustrated in Figure 1, suppose there are *N* edge devices in an edge network. Regarding the available memory, computation capacity, and energy, various edge devices can perform different sub-tasks of a given inference. Then, when an intelligent edge application needs to be deployed, the corresponding edge server will act as a master to break down the DNN inference tasks according to the current states of all edge devices and the communication bandwidth of the edge network. Similar to [6], this process can be divided into three main stages, as shown in Figure 2.

At the offline training stage, a BranchyNet is trained for running. Then, the online partitioning and deploying stage will first predict the performance of each layer in the pre-trained model. This can be achieved by PALEO [33], the layer performance regression function in [6,18], or other layer prediction models. Based on the result of layer performance prediction, the model partitioning algorithm generates an optimal deployment, according to which the sub-tasks are assigned to selected edge devices. Then, these edge devices perform the original inference task collaboratively at run-time. Unlike offline partitioning, which partitions a model immediately after training without considering existing available resources in the run-time environment, this work focuses on online partitioning that tries to achieve better execution performance in a given run-time environment.

In real-world applications, the online partitioning and deploying algorithm will be deployed on the edge server. Then, after a user deploys an intelligent application to the edge environment, the edge server will extract the inferencing task and the corresponding BranchyNet. It partitions and dispatches the BranchyNet according to the current status of each edge device which will then cooperate to complete a further distributed inferencing process without the edge server.

### 3.2. Duration for Performing a Sub-Task in the Cooperative Inferencing

Each device must receive input from its preceding device and deliver the output to its next device during cooperative inferencing. For example, in Figure 2, the B-AlexNet is divided into five sub-models and deployed to five different devices. To perform the assigned inference task, the camera needs to receive input from the first drone and send its output to the smartphone, which will, in turn, execute the next sub-model.

Let us suppose a complete inference task is divided into *n* sub-tasks deploying to *n* selected devices. This paper uses $p_{i}\left(i=1,2,\ldots ,n\right)$ to represent a sub-model to be performed and $d_{j}\left(j=1,2,\ldots ,n\right)$ to represent one of those selected devices. If a sub-model $p_{i}$ is deployed to device $d_{j}$, the corresponding execution time $t_{i,j}$ is defined as
(1)$$t_{i,j}=tc_{i,j}+tr_{i}+ts_{i}$$
where $tc_{i,j}$ is the time of executing sub-model $p_{i}$ on device $d_{j}$, $tr_{i}$ and $ts_{i}$ are the time for receiving the input of $p_{i}$ and sending the output of $p_{i}$, respectively. If $tt_{i}$ is used to represent the total transmission time, then $tt_{i}=tr_{i}+ts_{i}$ and $t_{i,j}=tc_{i,j}+tt_{i}$.

### 3.3. Duration for Executing a Complete BranchyNet

Different from conventional linear structure DNNs, a BranchyNet usually has several early-exit branches. Based on the idea of an early-exit mechanism, not all the branches would be executed for processing a given input. Therefore, it is different from the total execution time calculation for linear structure DNNs, which can be easily achieved by summing up the execution duration of all the sub-tasks. The total execution time of a BranchyNet varies according to the complexity of the given input. This paper adopts the weighted average duration to represent the duration of performing a complete BranchyNet inference.

In detail, suppose there are *m* branches in a given BranchyNet. Each branch $b_{i}(i=1$, 2,…,m) has a total execution duration $tb_{i}$ and a proportion representing the percentage of samples exit from this branch $pb_{i}$. Then, the duration for performing the given BranchyNet is defined as
(2)$$T=\sum_{i=0}^{m}pb_{i}\times tb_{i}$$
where $\sum_{i=0}^{m}pb_{i}=1$. The value of each $pb_{i}$ can be achieved after training or testing the BranchyNet model, which equals the ratio of the number of samples exiting from branch $b_{i}$ and the total number of samples, while the execution time of a complete branch $b_{i}$ is obtained by summing up the execution time of all layers that a sample traverses before exiting, i.e.,
(3)$$tb_{i}=\sum_{i=1}^{n}\sum_{j=1}^{n}\alpha_{i,j}\times t_{i,j}$$
where $\alpha_{i,j}$ is a coefficient to indicate whether a sub-model $p_{i}$ is assigned to device $d_{j}$, whose value is either 0 or 1. When $\alpha_{i,j}=1$, the sub-model $p_{i}$ is assigned to device $d_{j}$. Otherwise, i.e., $\alpha_{i,j}=0$, the sub-model $p_{i}$ is assigned to another device rather than $d_{j}$. If a branch is divided into *n* parts, and each part is uniquely deployed to one specific device, then, for any device $d_{j}\left(j=1,2,\ldots ,n\right)$, the equation $\sum_{i=1}^{n}\alpha_{i,j}=1$ makes sense. Likewise, for any sub-model $p_{i}\left(i=1,2,\ldots ,n\right)$, $\sum_{j=1}^{n}\alpha_{i,j}=1$.

### 3.4. Problem Formulation

Except for the above time cost for performing a BranchyNet, given the memory and power consumption requirements of performing each sub-model, it is also necessary to consider the available amount of memory and energy on each device to achieve a feasible partition. In summary, this work aims to minimize the execution time of a given BranchyNet under the constraints of memory and energy consumption, which is formulated the BranchyNet partition problem as the following constrained optimization problem.
(4)$$\begin{array}{cc} \; & \min T\\ \; & s.t.\;\left\{\begin{array}{cc} rm_{i}\leq m_{j}, & \alpha_{i,j}=1,\\ ep_{j}\times t_{i,j}\leq c_{j}, & \alpha_{i,j}=1. \end{array}\right. \end{array}$$

Here, $m_{j}$ is the size of available memory on device $d_{j}$, and $rm_{i}$ is the required memory for running sub-model $p_{i}$. If $p_{i}$ is assigned to device $d_{j}$, i.e., $\alpha_{i,j}=1$, the available amount of memory $d_{j}$ should not be less than the required amount of memory $rm_{i}$. Similarly, the remaining energy $c_{j}$ should not be less than the required amount of energy, which is the product of the average running power $ep_{j}$ and the sub-model execution time $t_{i,j}$. If there are more constraints, such as computing capabilities, communication bandwidth, and other aspects, the problem can be modeled by adding corresponding constraints to the above formulation.

Based on Equations (Equation 2) and (Equation 3), the problem can be finally defined as follows:(5)$$\begin{array}{cc} \; & \min\sum_{k=0}^{m}\left(pb_{i}\times \sum_{i=1}^{n_{bk}}\sum_{j=1}^{n_{bk}}\alpha_{i,j}\times t_{i,j}\right)\\ \; & s.t.\;\left\{\begin{array}{cc} rm_{i}\leq m_{j}, & \alpha_{i,j}=1,\\ ep_{j}\times t_{i,j}\leq c_{j}, & \alpha_{i,j}=1. \end{array}\right. \end{array}$$
where $n_{bk}$ is the number of pieces branch $b_{k}$ is divided into.

All of these $\alpha_{i,j}\left(\forall i,j\in \left\{1,\ldots ,n\right\}\right)$ form an n-by-n matrix *A*, which represents an actual deployment plan. This paper aims to achieve a specific matrix *A* with the shortest execution time under memory and energy consumption constraints. Let us suppose a BranchyNet with *L* layers will be partitioned and deployed to *N* devices. There are $C_{L-1}^{N-1}$ different partition plans and $P_{N}^{N}$ different deployment plans. Therefore, the above problem is a typical NP problem.

## 4. The Proposed Two-Layer Chromosome GA

A genetic algorithm (GA) is a method to search for the optimal solution by simulating the natural evolution process. When solving complex combinatorial optimization problems, a GA can usually obtain better optimization results faster than some conventional optimization algorithms. The chromosome coding scheme is the fundamental element of a GA. This section analyzes problems of applying the basic GA to solve the formulated optimization problem and then introduces the corresponding solution in this paper.

### 4.1. Linearization of a BranchyNet

In order to calculate and represent the partitioning plan conveniently, a linearization strategy is proposed to convert a BranchyNet into a linear model. The linearization result of the B-AlexNet is shown in Figure 3.

In Figure 3, the first line illustrates the linearized results of the B-AlexNet, where circles are the fork nodes in the B-AlexNet. To distinguish different layers in a B-AlexNet, this work labels a layer as $l_{x,y}$, where *x* refers to the branch the layer belongs to, and *y* indicates the position that the layer is located in the branch. For example, in the B-AlexNet, as there are three branches, the value of *x* is 0, 1, or 2. $l_{1,2}$ is the second layer of the first exit branch. To simplify the representation and computation in the proposed algorithm, layers in the linearized model are further transformed to $L_{i}\left(i=1,2,\ldots ,n\right)$, and a mapping is established between labels in the linearized model layer and their original position.

After the above linearization, a partition plan can be easily represented as a vector. As shown in the middle of Figure 3, if there are *L* layers in the linearized model, there should be L−1 possible cut points. Hence, a partitioning plan can be modeled as a vector with L−1 elements, each of which is either 0 or 1, and the value 1 represents a selected cut point. The B-AlexNet is divided into five parts in the illustrating example, so there are four cut points, as shown in Figure 3.

Moreover, a deployment plan can be modeled as an N×L matrix, *A*. The value of any $a_{i,j}$ in *A* is either 1 or 0. If $a_{i,j}=1$, it means that layer $L_{j}$ is assigned to device $d_{i}$. As shown at the bottom of Figure 3, there are five devices and thirteen layers. A deployment plan is modeled as a matrix with five rows and thirteen columns. The value 1 at the first line means layers $L_{1}$, $L_{2}$, $L_{3}$, and $L_{4}$ are deployed to device $d_{1}$, where $L_{1}$, $L_{2}$, $L_{3}$, and $L_{4}$ are the identifiers of layers in the linearized model.

### 4.2. Problems with Basic Crossover Operator

The basic process of a GA starts with generating an initial population, i.e., a set of chromosomes. Then, it runs through the loop, including individual evaluation, selection, crossover, and mutation, until satisfying the given termination condition. The crossover operator plays a core role in a GA, which acts on a group of chromosomes and generates new individuals by replacing part of the chromosomes of two father-generation individuals.

For example, in order to obtain an optimal deployment plan, the chromosome in the GA should be modeled as an N×L matrix according to the above representation of a deployment plan, such as $C_{1}$ and $C_{2}$ in Figure 4. $C_{1}$ represents a deployment plan where there are three devices and a seven-layer model. The model is divided into three pieces. Layers $L_{1}$ and $L_{2}$ are deployed to device $d_{1}$, layers $L_{3}$, $L_{4}$, and $L_{5}$ are deployed to device $d_{2}$, and layers $L_{6}$ and $L_{7}$ are deployed to device $d_{3}$. Similarly, $C_{2}$ also shows a deployment plan which divides the model into three pieces. Figure 4 shows the computing process in a partially mapped crossover operator.

In Figure 4, $C_{11}$ and $C_{21}$ are two new individuals generated by swapping the subsections in each father individual included in the rectangles. According to the chromosome coding rules, the model is divided into five parts in the newly generated individuals, $C_{11}$ and $C_{21}$. In other words, a five-part partitioning plan is generated by the crossover operation based on two three-part partitioning plans.

Furthermore, such deployment plans will lead to extra network bandwidth and equipment energy consumption caused by repeated transmission between devices. For example, if deploying the inference task according to $C_{11}$, the output of $L_{1}$ will be sent from $d_{1}$ to $d_{2}$, and then the output of $L_{2}$ will be sent back from $d_{2}$ to $d_{1}$. In turn, the output of $L_{4}$ will be sent from $d_{1}$ to $d_{2}$ again. As a result, the intermediate results need to be transferred four times among the given three devices, twice as many as deployed according to $C_{1}$.

### 4.3. The Proposed Improvement

To ensure the new individuals generated by crossover are still consistent with the corresponding partitioning plans their parents belong to, this work proposes dividing the chromosomes into two classes, i.e., partitioning chromosomes and deployment chromosomes. Based on the representation introduced above, if an *L*-layers BranchyNet will be partitioned and deployed to *N* devices, this work adopts the representation of a partitioning plan which is a vector of length L−1 as a partitioning chromosome and the representation of a deployment plan which is an N×L matrix as a deployment chromosome.

As illustrated in Figure 3, there is a one-to-many relationship between the partitioning chromosome and the deployment chromosome. Let us suppose there are *N* devices to participate in the collaborative inferencing, and each device will be only assigned one sub-model. Then, the total number of deployment chromosomes related to a given partitioning chromosome is N!. Conversely, only one partitioning chromosome can be abstracted from a given deployment chromosome. Based on the relationship between the partitioning and deployment chromosomes, it is easy to build up corresponding conversion algorithms.

Then, this work modifies the process in the basic GA by performing crossover and mutation on partitioning chromosomes and selection on deployment chromosomes. A complete process of the improved GA is shown in Algorithm 1.
**Algorithm 1:** The framework of the proposed genetic algorithm
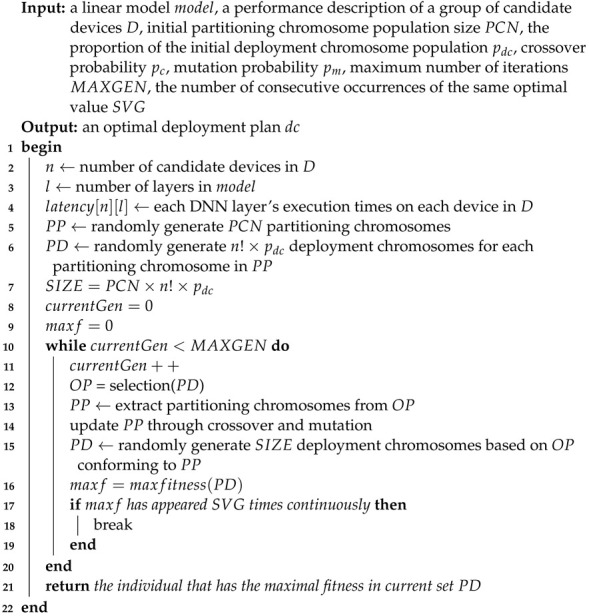


Firstly, Algorithm 1 takes a linearized model model to be partitioned and its layer-performance estimation on every candidate device *D* as input. Moreover, since the algorithm constructs two layers of populations, i.e., the partitioning population and the deployment population, the initial sizes of these two populations need to be set. As shown in the input description, this algorithm takes an input parameter PCN to configure the number of initial individuals in the partitioning population. Then, if there are *n* devices to participate in the collaborated inferencing, there will be PCN×n! possible deployment individuals. To control the size of the deployment population, the algorithm adopts an additional parameter, $p_{dc}\left(0<p_{dc}<=1\right)$, which is a proportion for generating deployment individuals. As a result, the size of the deployment population can reduce to $PCN\times n!\times p_{dc}$.

Then, the algorithm can be divided into two parts. The first section consists of lines 2 through 9 and is responsible for initializing the algorithm. The second section lines 10 through 20, is the core process of the algorithm. In each iteration of the genetic algorithm, an optimal group of individuals OP is selected from the current deployment population PD (line 12), and the corresponding partitioning chromosome is extracted (line 13). The algorithm performs crossover and mutation operations on these partitioning chromosomes and generates a new generation of partitioning chromosomes (line 14). Then, based on the newly generated partitioning population, the existing excellent deployment chromosomes are filtered, and some new deployment chromosomes will be generated at the same time to complete the iteration of the deployment population (line 15). Finally, the current maximum fitness value is calculated on the newly generated deployment population, and the loop is terminated when the termination condition is met (lines 16 to 19).

The time complexity of GA is O(TNM), where *T* is the maximum iteration times, *N* is the population size, and *M* is the number of genes in each individual. As we set the maximum iteration time to 200 in our experiments, the time complexity is O(200NM). The corresponding codebase is open-source and can be found at https://github.com/handuoZhang/PGA (accessed on 31 December 2022).

According to the problem formulation described above, the fitness function is defined as follows:(6)$$\mathrm{fitness}\left(dc\right)=\left\{\begin{array}{cc} \frac{10^{5}}{\sum_{k=0}^{m}\left(pb_{i}\times \sum_{i=1}^{n_{bk}}\sum_{j=1}^{n_{bk}}dc_{i,j}\times t_{i,j}\right)} & dc\;\mathrm{satisfies}\;\mathrm{all}\;\mathrm{constraints}\\ 10^{-6} & dc\;\mathrm{does}\;\mathrm{not}\;\mathrm{satisfy}\;\mathrm{all}\;\mathrm{constraints} \end{array}\right.$$
Here, dc is the α matrix in Equation (Equation 5), representing a specific deployment plan.

## 5. Performance Evaluation

This section demonstrates the effects of the proposed algorithm. To provide a comprehensive comparison, this work evaluates the performance of the proposed BranchyNet partitioning approach on the B-AlexNet and B-ResNet proposed in [1] compared with four other typical DNN partitioning methods. Model training and inferencing are performed on the Cifar-10 dataset [34], which consists of 32×32 color images divided into ten classes. Furthermore, all of the experimental results are collected by running the same algorithm ten times as a group on a laptop with an AMD Ryzen7 5700U CPU and 16 GB memory in a Pycharm environment.

### 5.1. Experiments on Partitioning B-AlexNet

Firstly, a simulated distributed system with six devices with different configurations is set up, as shown in Table 1.

As mentioned above, PALEO [33] is a performance model that can estimate DNN performance under a given deployment assumption. Therefore, this experiment employs PALEO to evaluate the memory requirements and execution time of different DNN layers running on each candidate device in Table 1 and construct corresponding performance description *D* in Algorithm 1.

Based on the number of devices to join into an inference, the initial partitioning and deployment chromosome population size are set as follows in Table 2:

The other parameter values in Algorithm 1 are set as follows. In both the basic GA and the improved GA, the crossover probability is 0.5, the mutation probability is 0.01, the maximum iteration number is 200, and the algorithm will be terminated when the optimal fitness value remains unchanged for 50 consecutive generations. In the basic GA, the chromosomes are modeled as deployment chromosomes.

To compare the resulting total execution time generated by different algorithms, this work considers two scenarios according to whether the partitioning and deploying are considered simultaneously.

#### 5.1.1. Total Execution Time Comparison under Considering Partitioning and Deploying Separately

Considering partitioning and deploying separately means that the partitioning plan and deployment plan are generated one after another. In other words, the optimization is divided into two steps, i.e., first, to generate an optimal partitioning plan and then generate an accordingly optimal deployment plan. Specifically, the following experiment compares the proposed algorithm with generating the optimal deployment plan based on exhaustive searching after partitioning the B-AlexNet by the algorithms proposed in [18,19] and partitioning the B-AlexNet on one of the two fork points.

Here, the network structure and performance settings in implementing the Neurosurgeon DNN segmentation algorithm [18] are the same as in the proposed GA introduced above. Moreover, the current data center load level is set to 0. Moreover, the local processing time, edge processing time, and output transmission time of each DNN layer in the shortest-path-based approach [19] are also calculated by PALEO, and the corresponding SDAG in this approach is constructed according to the method in [19]. The settings in other experiments are the same as those in this experiment, which will not be described below. Figure 5 shows the average total execution time and device-average energy cost.

The above results show that the proposed GA results in a shorter average total execution time and lower device-average energy cost than other approaches. The reason is that the improved GA tries to find the best partitioning plan based on the corresponding performance when actually deploying to a given environment. A more detailed comparison of total execution time is shown in Table 3.

As shown in Table 3, the average total execution time resulting from the proposed algorithm is shorter than other approaches. Moreover, the standard deviation is almost 0, which indicates that the proposed algorithm is also more stable.

#### 5.1.2. Total Execution Time Comparison When Considering Partitioning and Deploying Simultaneously

As a GA is an approximate algorithm, it cannot ensure obtaining the absolute optimal solution. This experiment adopts an exhaustive method that can provide the optimal total execution time and energy cost in any given setting as the baseline and then compares the corresponding results by running the basic and improved GA. The following figure shows the average total execution time and device-average energy cost by dividing the B-AlexNet into different pieces from three to six.

As shown in Figure 6, both the average total execution time and the device-average energy cost of the proposed GA are close to the absolute optimal value, which is visibly superior to a basic GA. Compared with the absolute optimal value, the proposed algorithm has a difference of about 3%, 8%, 21%, and 34% in average total execution, and a difference of about 2%, 3%, 12%, and 23% in the device-average energy cost, respectively. A more detailed comparison of the total execution time is shown in Table 4.

Here, the maximum and minimum values of total execution time obtained from the actual experiment show the variation range of all resulting total execution times. The average and median values reflect the most common values of all resulting total execution time, and the standard deviation indicates their volatility. From the above statistical results, it can be seen that the gap between the improved GA and the optimal value increases with the complexity of the problem. When deploying an *L*-layers BranchyNet to *N* devices, the total number of possible deployment plans is $N!\times C_{L}^{N}$, i.e., L!/(L−N)!. For example, in this experiment, when the number of partitions reaches 6, there are 6!, i.e., 720, possible deployment plans for each partitioning plan. If each partition contains at least one convolutional layer, there are $C_{8}^{6}$, i.e., 28, possible partitioning plans. As a result, the number of possible deployment plans reaches $6!\times C_{8}^{6}$, i.e., 20,160. This is why there is a sudden decrease in algorithm stability. However, compared with the basic GA, the proposed GA achieves a shorter total execution time which is less than half of that of the basic GA.

Moreover, this work compares the average computation time for generating a deployment plan by exhaustive searching, basic GA, and the proposed GA. Table 5 shows the results. As mentioned above, the primary and improved GA will stop when an optimal fitness value remains unchanged for 50 consecutive generations or they reach the maximum iteration number of 200 in the experiment setting. Moreover, all the algorithms are executed ten times, and the average value is calculated and compared in the following table.

In Table 5, the rightmost column shows the execution time of the exhaustive searching compared to the proposed GA, which increases with the number of partitions rapidly. When the number of partitions is relatively small, the basic GA and proposed GA take more time to satisfy the terminal condition, which is longer than the time needed by exhaustive searching. However, as the number of partitions increases, the execution time required by the exhaustive method increases explosively, which is significantly higher than that of both GAs.

### 5.2. Experiments on Partitioning B-ResNet

Similar to the experiments above, the experiments setting and results on partitioning B-ResNet are as follows.

As the number of layers in a B-ResNet is much larger than in a B-AlexNet, this experiment constructs a simulated distributed system with only four different devices. Table 6 shows the device configurations, and Table 7 shows the initial population size. Other parameter settings are the same as those in the experiments on B-AlexNet.

In the scenario that considers partitioning and deploying separately, this experiment also compares the proposed GA with the approaches partitioning the given B-ResNet in two parts and then selecting an optimal deployment plan. Figure 7 shows the comparisons of optimal average total execution time and device-average energy cost calculated by each method. Moreover, Table 8 shows the comparison of statistical results of the resulting total execution time.

From the above results, it can be found that the proposed GA also has the best stability from the other methods and can result in a shorter total execution time. However, the base model of B-ResNet is ResNet-110 [1], which has 109 convolutional layers and 1 fully-connected layer. Only adding two branches that totally have 5 convolutional layers and 2 fully-connected layers increases less than 5% computation, which has little impact on the overall execution time and energy cost. This is the reason why the proposed GA in this experiment is not as effective as the previous experiment.

In the scenario that considers partitioning and deploying simultaneously, this experiment also compares the average total execution time and device-average energy cost of the proposed GA with the optimal value and that of the basic GA, which is shown in Figure 8. Moreover, the statistical results of the resulting average total execution time are compared in Table 9.

Similar to the corresponding results in experiments on B-AlexNet, the proposed GA also presents higher stability than the basic GA. However, as the number of possible deployment plans increases dramatically in partitioning a B-ResNet, the differences between the results of the proposed GA and the optimal value achieved by exhaustive searching turn out to be larger than that of partitioning B-AlexNet. However, the algorithm can be fine-tuned to achieve a better result by adjusting its maximum number of iterations and terminal condition.

At last, Table 10 shows the comparison on execution time by running different algorithms.

The rightmost column in Table 10 shows how many times the execution time of the exhaustive searching compared to the proposed GA, which increases with the number of partitions rapidly. Similar to the conclusion in partitioning B-AlexNet, as the number of partitions increases, the execution time required by the exhaustive method increases explosively, which is significantly higher than that of the two GAs, and the proposed GA shows its advantage in applying it for performing online partitioning.

## 6. Conclusions

This paper proposes a GA-based BranchyNet partitioning approach for accelerating edge inference. Considering the structural particularity of BranchyNet, this paper puts forward a weighted-average calculation approach for estimating the BranchyNet total execution time. Moreover, it proposes a two-layer chromosome GA by distinguishing partition and deployment during the evolution of a GA. In detail, there are two evolution levels in the proposed GA. On one side, crossover and mutation perform on partitioning chromosomes, ensuring the top-down consistency between the partitioning population and deployment population. On the other side, selection performs on the deployment chromosomes, which further drives the evolution of the partitioning population by ensuring down–top consistency.

In order to show the effects of the proposed approach, this work conducts a group of comprehensive experiments on both B-AlexNet and B-ResNet. The experiment results show that the proposed algorithm can not only result in shorter inferencing time and lower device average energy cost but also requires less time to obtain an optimal deployment plan. Such short running time of the proposed algorithm enables it to generate an optimal deployment plan online to satisfy the actual requirements in deploying an intelligent application dynamically. To further improve this work, the approach for finding the best settings of the algorithm parameters needs to be further studied to obtain better operation effects.

## Figures and Tables

**Figure 1 sensors-23-01500-f001:**
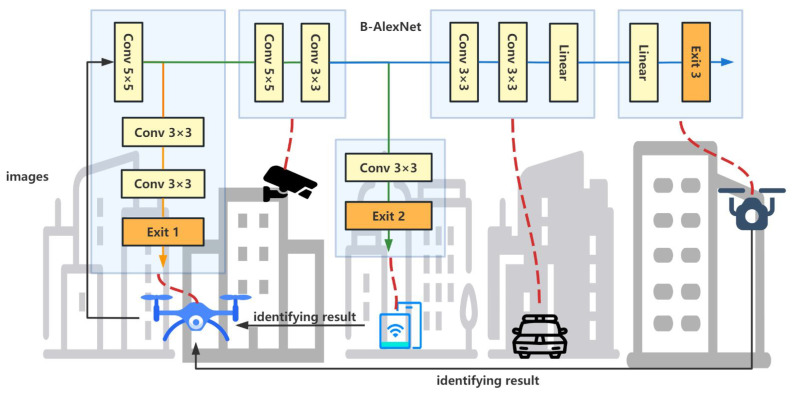
Illustration of a distributed DNN inference by collaboration among multiple edge devices.

**Figure 2 sensors-23-01500-f002:**
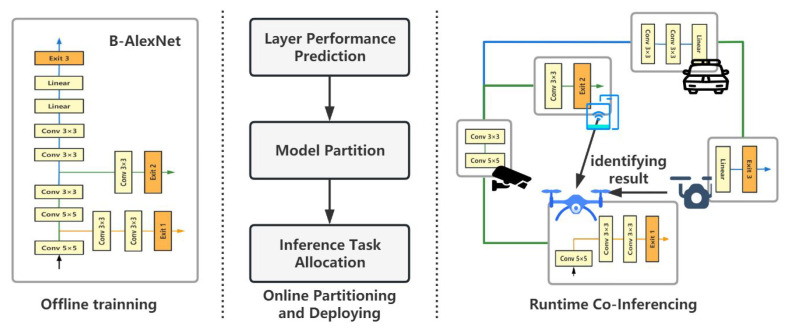
The basic process for partitioning and distributing inference tasks contained in an intelligent edge application.

**Figure 3 sensors-23-01500-f003:**
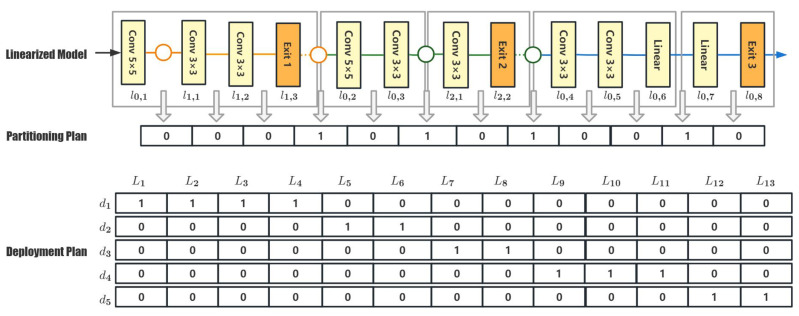
An example of relationship among a linearized BranchyNet, a partitioning plan, and a deployment plan.

**Figure 4 sensors-23-01500-f004:**
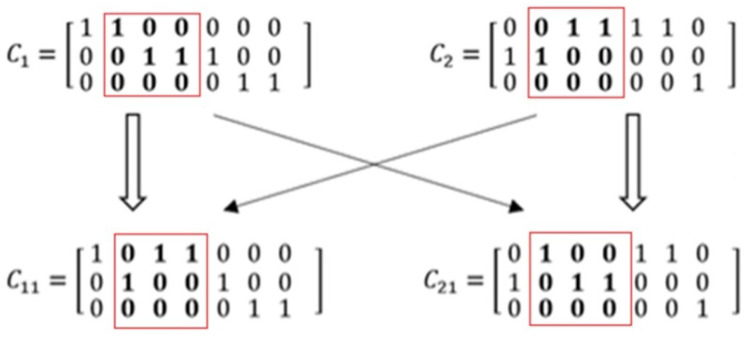
Illustration of the computing process in the partially mapped crossover operator.

**Figure 5 sensors-23-01500-f005:**
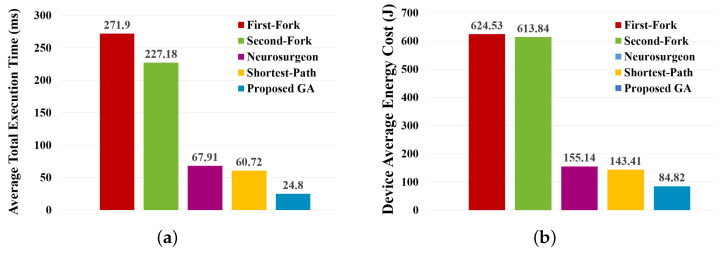
Performance comparison on partitioning B-AlexNet by considering partitioning and deploying separately. (**a**) Comparison on average total execution time. (**b**) Comparison on device-average energy cost.

**Figure 6 sensors-23-01500-f006:**
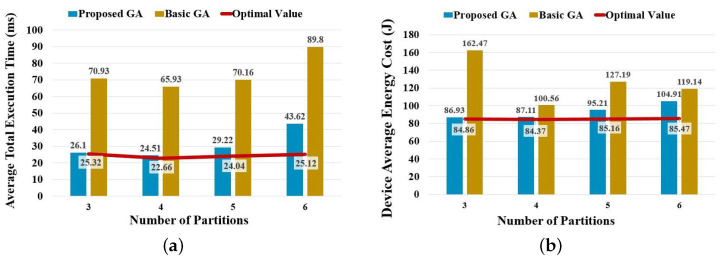
Performance comparison on partitioning B-AlexNet by considering partitioning and deploying simultaneously. (**a**) Comparison on average total execution time. (**b**) Comparison on device-average energy cost.

**Figure 7 sensors-23-01500-f007:**
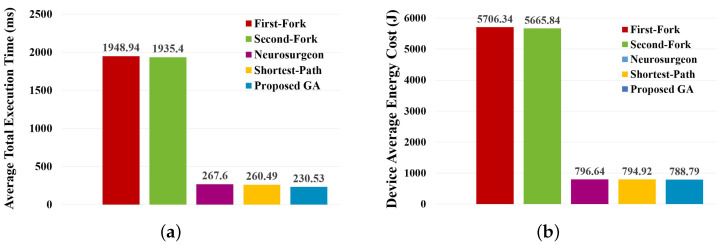
Performance comparison on partitioning B-ResNet by considering partitioning and deploying separately. (**a**) Comparison on average total execution time. (**b**) Comparison on device-average energy cost.

**Figure 8 sensors-23-01500-f008:**
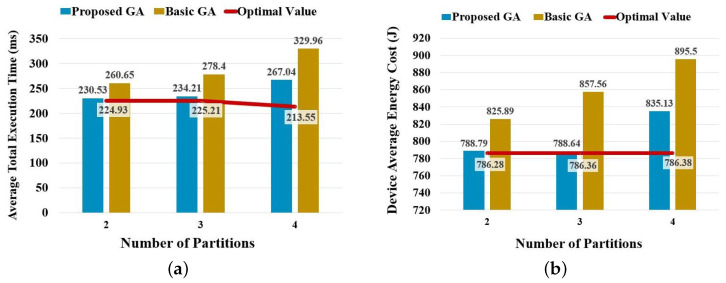
Performance comparison on partitioning B-ResNet by considering partitioning and deploying simultaneously. (**a**) Comparison on average total execution time. (**b**) Comparison on device-average energy cost.

**Table 1 sensors-23-01500-t001:** Performance parameters of edge devices in experiments on B-AlexNet.

Device No.	GFLOPS	I/O Bandwidth (MBPS)
1	0.218	140.85
2	9.920	1525.63
3	0.213	135.89
4	13.500	1698.25
5	0.247	140.91
6	3.620	159.45

**Table 2 sensors-23-01500-t002:** The initial partitioning and deployment chromosome population size in experiments on B-AlexNet.

Number of Devices	Initial Size of Partitioning Population	Initial Size of Partitioning Population	Proportion of the Initial Deployment Chromosomes
2	2	4	1
3	2	12	1
4	2	48	1
5	2	120	0.5
6	2	144	0.1

**Table 3 sensors-23-01500-t003:** Comparison of B-AlexNet total execution time (ms) found by different approaches.

Approach	Average	Maximum	Minimum	Mode	Standard Deviation
First-Fork	271.90	315.48	206.53	315.48	56.26
Second-Fork	227.18	237.87	202.23	237.87	17.22
Neurosurgeon [18]	67.91	96.67	24.76	96.67	37.13
Shortest-Path [19]	60.72	96.67	24.76	24.76	37.90
Proposed GA	24.80	24.86	24.76	24.81	0.03

**Table 4 sensors-23-01500-t004:** Comparison of B-AlexNet total execution time in deploying to from three to six devices.

Number of Partitions	Approach	Average	Maximum	Minimum	Mode	Standard Deviation
3	Opt. Value	25.32	25.32	25.32	25.32	0
	Basic GA	70.93	100.78	25.32	97.48	36.11
	Prop. GA	26.10	32.79	25.32	25.38	2.35
4	Opt. Value	22.66	22.66	22.66	22.66	0
	Basic GA	65.93	98.75	23.48	61.65	28.41
	Prop. GA	24.51	30.36	22.90	22.90	2.42
5	Opt. Value	24.04	24.04	24.04	24.04	0
	Basic GA	70.16	97.16	26.85	50.19	25.27
	Prop. GA	29.22	34.86	24.05	30.82	3.76
6	Opt. Value	25.12	25.12	25.12	25.12	0
	Basic GA	89.80	153.49	36.79	66.08	33.82
	Prop. GA	43.62	86.87	30.86	38.57	16.32

**Table 5 sensors-23-01500-t005:** Comparison of the average computation time for generating a deployment plan of a given B-AlexNet.

Number of Partitions	Exhaustive Searching (ms)	Basic GA (ms)	Proposed GA (ms)	ES/PGA
3	577.37	630.46	575.29	1
4	3111.94	3387.30	2395.67	1.3
5	127,744.53	9193.03	8207.73	15.6
6	5,612,393.62	28,999.22	19,932.17	131

**Table 6 sensors-23-01500-t006:** Performance parameters of edge devices in experiments on B-ResNet.

Device No.	GFLOPS	I/O Bandwidth (MBPS)
1	0.218	140.85
2	9.920	1525.63
3	0.213	135.89
4	13.500	1698.25

**Table 7 sensors-23-01500-t007:** The initial partitioning and deployment chromosome population size in experiments on B-ResNet.

Number of Devices	Initial Size of Partitioning Population	Initial Size of Partitioning Population	Proportion of the Initial Deployment Chromosomes
2	2	4	1
3	2	12	1
4	2	48	1

**Table 8 sensors-23-01500-t008:** Comparison of B-ResNet total execution (ms) resulted by different approaches.

Approach	Average	Maximum	Minimum	Mode	Standard Deviation
First-Fork	1948.94	3588.72	309.15	309.15	1728.496
Second-Fork	1935.40	2016.63	1881.24	1881.24	69.92
Neurosurgeon [18]	267.60	296.04	224.93	296.04	36.72
Shortest-Path [19]	260.49	296.04	224.93	224.93	37.48
Proposed GA	230.53	243.55	224.93	224.96	8.98

**Table 9 sensors-23-01500-t009:** Comparison of B-AlexNet total execution time in deploying to three to six devices.

Number of Partitions	Approach	Average	Maximum	Minimum	Mode	Standard Deviation
2	Opt. Value	224.93	224.93	224.93	224.93	0
	Basic GA	260.65	313.56	224.93	241.92	30.35
	Prop. GA	230.53	243.55	224.93	224.96	8.98
3	Opt. Value	225.21	225.21	225.21	225.21	0
	Basic GA	278.40	312.78	242.21	296.47	26.67
	Prop. GA	234.21	243.91	225.21	225.22	9.50
4	Opt. Value	213.55	213.55	213.55	213.55	0
	Basic GA	329.96	373.71	235.17	339.48	40.32
	Prop. GA	267.04	305.18	230.53	277.57	26.58

**Table 10 sensors-23-01500-t010:** Comparison of the average computation time for generating a deployment plan of a given B-ResNet.

Number of Partitions	Exhaustive Searching (ms)	Basic GA (ms)	Proposed GA (ms)	ES/PGA
2	534.76	572.74	558.87	1
3	22,076.23	3904.00	3656.95	6
4	9,549,783.88	10,590.18	8878.47	1075.6

## Data Availability

The datasets can be obtained from http://www.cs.toronto.edu/~kriz/cifar.html (accessed on 17 November 2021).

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
