# Peer review of "Genetic Algorithm-Based Online-Partitioning BranchyNet for Accelerating Edge Inference"

_sensors, 2023, doi:10.3390/s23031500_

Round 1

Reviewer 1 Report

Comments to authors:

Authors proposed genetic algorithm to edge inference accelerating with early exit scheme. Experiments demonstrate the effectiveness of the proposed scheme.

Advantages:

--A problem formulated to optimize the task completion time with energy and storage constraints.

--Simulation experiments demonstrate the performance of proposed scheme.

Drawbacks:

--For problem formulation, only energy and memory constraints have been included. However, the computing capability and the channel bandwidth have not been addressed.

--It is a pity that the difficulty of the formulated problem has not been discussed. Is it NP-hard? Is it convex? Is it non-convex? 

--Only one application with early exit has been addressed. However, in the real world, there are multiple heterogeneous applications, how to make sure the proposed scheme can adapt to multiple heterogeneous applications with different application finishing time constraints?

--It is not convincing when using Genetic algorithm to compare with existing works. Because the parameters for genetic algorithm greatly affect the performance of the scheme. How to decide the optimal parameter value should be discussed. Meanwhile, the parameters used in this work may differ from existing works, meanwhile, the parameters in different benchmark algorithms may also be different. Therefore, how to guarantee a fair comparison? More detailed should be given.

--The application scenario should be better demonstrated. In real world application, how to use the proposed scheme to optimize the performance of what applications?

--The references are not new. Authors are recommended to include recent works in the year 2022.

--Why not compare the performance with distributed scheme in reference [24]?

--Where can the algorithm be deployed? 

--The scheme seems a centralized scheme, what's the advantages over distributed schemes, e.g., [24] in the manuscript?

--The convergence of the proposed scheme should be proved.

--The time complexity of the proposed scheme should be given.

--Why not give detailed solutions using conventional algorithm?

Author Response

Dear reviewer,

Thanks very much for taking the time to review our manuscript. I really appreciate all your comments and suggestions! I have revised the problems you proposed one by one and the details are as follows:

(1) Any DNN solving a classification problem can be resized into the corresponding BranchyNet version. The algorithm proposed in this paper is mainly applied to distribute a BranchyNet model to a given group of edge devices. By fully using the edge resources, the response time of intelligent applications is shortened, and the network bandwidth is saved. In addition to the public security system and city surveillance system mentioned in the paper, there are also many other examples of applications in UAV collaboration, vehicle networking, and smart home systems. In the introduction part, we have given a detailed description of an example application to help readers find more similar real-world application scenarios. In addition, since BranchyNet generally refers to any neural network with one or more early exit points, the method proposed in this paper is not just designed for a specific application or neural network. It is applicable to partition any given BranchyNet model. Section 3.1 showed the basic process for partitioning and distributing any intelligent application. Besides, the proposed GA can partition any given BranchyNet. To clarify, we added a reference to our prior work, which contains a fall detection application example, and another reference to the work of traffic prediction at line 73 and line 74.

(2) Although only the energy and memory constraints are considered in the problem formulation, the computing capability and channel bandwidth you listed, as well as any other constraints, can be treated in the same way. Moreover, the influence of communication bandwidth is also considered in the calculation of cooperative inferencing delay now. To clarify, we added relevant descriptions from line 220 to line 223.

(3) The proposed algorithm will be deployed on the edge server. Then, after a user deploys an intelligent application to the edge environment, the edge server will extract the inferencing task and the corresponding BranchyNet. It partitions and dispatches the BranchyNet according to the current status of each edge device which will then cooperate to complete a further distributed inferencing process without the edge server. We have added the necessary description from line 166 to 171 to make the description more precise.

(4) So, our algorithm generates a distributed deployment plan of a given BranchyNet, which is similar to the aim of partitioning in reference 26 (originally 24). However, different from partitioning a liner DNN in reference 26, we try to partition a BranchyNet, which is not a liner structure. Therefore, we proposed a new approach to calculate the total inference time and formulate the corresponding optimization objective function. In contrast, the method in reference 26 cannot be applied on BranchyNet directly. That's the reason why we didn't provide the comparison.

(5) Similar to the partition problem formulation, the problem in this paper is also non-convex. Moreover, the time complexity of GA is O(TNM), where T is the maximum iteration times, N is the population size, and M is the number of genes in each individual. As we set the maximum iteration time to 200 in our experiments, the time complexity is O(200NM). Please find corresponding descriptions from lines 228 to 229 and line 326 to line328.

(6) We have added two references published in 2022, i.e., reference 8 and 9.

The file named "diff.tex" marks up all my revision details. Thank you again for your valuable comments.

Reviewer 2 Report

This is a slightly quick review as I only have a week to finish it where I have not managed to be in the office even one day.

This paper is about using a genetic algorithm to split a neural network into multiple components which can be deployed to different devices which can then collaborate on the inference tasks.  The question is then how to perform the split to achieve the best performance or lowest energy consumption

Overall the paper is good quality and well written.  However, it lacks a reference to the code: while the description of the algorithm (Algorithm 1) is better than what we usually see in computing papers, there is almost no excuse to not publish code.  The tiny CIFAR dataset being referenced at the end is not sufficient.

In my admittedly limited experience with GAs, their results can be a bit hit and miss depending on starting conditions (sometimes) or how the chromosomes explore the solution space.  For the reader to verify that the PGA works as advertised, they should have an opportunity to download the code and try it out.  In particular, the CIFAR-10 dataset is a starting point but is very limited.

Specific comments

Starting from the back, in Table 5 (and 9), the indicated variables may not be the best way to describe the distribution as it looks skewed.  It may be worth plotting a few of the distributions to see if there are better ways to describe them.

In table 4 (and 10), the runtime of the GA is less relevant if it does not deliver good results compared to exhaustive search, and in fact you may need to run it several times.  The caption says "time to generate an optimal deployment plan" but how do you know it is optimal when only exhaustive search is known to give an optimal plan and GAs can give a spread of values (cf Tables 5/9).  It looks like Tables 5/9 are a bit misleading in this context.

Another thing which is not clear is whether the performance tests take into account cost and latency of communicating between the devices/partitions.  Judging from the flat "optimal" plot in Figs. 6/8, it would seem it is not taken into account, yet in the described scenario (devices communicating with each other) it matters a lot.

The follow up point is that optimisation for deployment (ie partitioning of the network into N partitions) depends on the scenario in which it is deployed.  Given that partitioning then needs doing only once, it is worth spending the computational effort to find a close-to-optimal solution, rather than achieving the partitioning in the shortest possible time.  After all, the network will spend most of its life doing inference, in which case the upfront cost of partitioning can be combined with the upfront cost of training it.

Line 185. You mean "pb_i"?

In summary. You should publish your code.

Author Response

Dear reviewer,

Thanks very much for taking the time to review our manuscript. I really appreciate all your comments and suggestions! First, as you suggested, we have posted the code on GitHub with the link "https://github.com/handuoZhang/PGA," and the link has been added to line 329 to 330. In addition, since the data sets used in training the based BranchyNet are MINST and CIFAR10 (in reference 1), we chose the same CIFAR10 data set for the convenience of using B-AlexNet and B-ResNet parameters that the author has trained and open source. Other details are revised as follows:

(1) According to Equations 1 and 5 in the problem formulation, the time cost of inference and communication was considered when calculating the collaborated inferencing delay. Therefore, the inference time in Figure 6 and Figure 8 includes the time of communication and inferencing. We have made a supplementary explanation from line 93 to 95 and changed all the “inference time” to “total execution time”.

(2) In Table 4 and Table 9, we aim to provide more detailed comparative data. We list the maximum and minimum values of inference time obtained from the actual experiment to show its variation range. The average and median values reflect the most common value, and the standard deviation indicates the volatility of resulting inference times. As a comparison of average inference times of B-AlexNet is already given in Figure 6(a), Table 4 is to provide more statistical comparisons. Similarly, the comparison of the average inference time of B-ResNet has been given in Figure 8(a), and Table 9 is to provide a more detailed comparison of its distribution. So the function of Table 4 and Table 9 is to highlight that the calculation results of our approach are closer to the optimal value with less fluctuation. To avoid ambiguity, we extended the above description in the paper. Please find from line 393 to line 396.

(3) Table 5 and Table 10 compare the average time to generate a deployment plan using different algorithms. Although the running time and results of the genetic algorithm have certain randomness, specific iteration termination conditions have been set in the experiment. So it is unnecessary to run many times to obtain a solution. As seen from Table 5 and Table 10, the actual inference time of a deployment plan generated by our approach is more stable and closer to the absolute optimal value. In contrast, its calculation time is much less than the exhaustive search algorithm generating the absolute optimal value. To clarify, we added relevant descriptions from line 409 to line 412 and changes to title to “Comparison of the average computation time for generating a deployment plan of a given B-AlexNet/B-ResNet”.

(4) Although DNN partitioning can be done immediately after training, we think there are various application scenarios. Traditionally, the model is partitioned according to its characteristics, such as model structures and parameter values. In this case, the model can be directly divided after training. However, this approach does not consider the limitations of available computing resources, such as device computing abilities and communication bandwidth, in the actual deployment environment. Thus, it cannot ensure a feasible deployment and achieve the optimal operational effect. On the contrary, this paper aims to generate a feasible deployment at run time when an application, including neural network inference computing, needs to be deployed in a distributed environment. An optimal deployment scheme is generated based on the state of available resources in the current deployment environment to ensure the feasibility of the deployment and the actual execution performance in the face of given network conditions. To clarify the paper, we added relevant descriptions from line 144 to line 146, and line 162 to line 165.

(5) We revised the mistake at line 198 by changing "bp_i" to "pb_i."

The file named "diff.tex" marks up all my revision details. Thank you again for your valuable comments.

Round 2

Reviewer 1 Report

The revised version shows some texts added. However, there are some critical issues.

(1) The similar problem has been well investigated by existing works, e.g.,

--Chen, J., Deng, Q., & Yang, X. (2023). Non-cooperative game algorithms for computation offloading in mobile edge computing environments. Journal of Parallel and Distributed Computing172, 18-31.

--Huang, X., Xu, K., Lai, C., Chen, Q., & Zhang, J. (2020). Energy-efficient offloading decision-making for mobile edge computing in vehicular networks. EURASIP Journal on Wireless Communications and Networking2020(1), 1-16.

--Sun, Y., Zhou, S., & Xu, J. (2017). EMM: Energy-aware mobility management for mobile edge computing in ultra dense networks. IEEE Journal on Selected Areas in Communications35(11), 2637-2646.

--Liu, H., Cao, L., Pei, T., Deng, Q., & Zhu, J. (2019). A fast algorithm for energy-saving offloading with reliability and latency requirements in multi-access edge computing. IEEE Access8, 151-161.

--Hou, X., Ren, Z., Wang, J., Cheng, W., Ren, Y., Chen, K. C., & Zhang, H. (2020). Reliable computation offloading for edge-computing-enabled software-defined IoV. IEEE Internet of Things Journal7(8), 7097-7111.

--Chen, Z., Chen, Z., & Jia, Y. (2019, December). Integrated task caching, computation offloading and resource allocation for mobile edge computing. In 2019 IEEE Global Communications Conference (GLOBECOM) (pp. 1-6). IEEE.

--Kuang, L., Gong, T., OuYang, S., Gao, H., & Deng, S. (2020). Offloading decision methods for multiple users with structured tasks in edge computing for smart cities. Future Generation Computer Systems105, 717-729.

--Chen, L., Wu, J., Zhang, J., Dai, H. N., Long, X., & Yao, M. (2020). Dependency-aware computation offloading for mobile edge computing with edge-cloud cooperation. IEEE Transactions on Cloud Computing.

--Lv, X., Du, H., & Ye, Q. (2022, May). TBTOA: A DAG-Based Task Offloading Scheme for Mobile Edge Computing. In ICC 2022-IEEE International Conference on Communications (pp. 4607-4612). IEEE.

--Li, M., Mao, N., Zheng, X., & Gadekallu, T. R. (2022). Computation offloading in edge computing based on deep reinforcement learning. In Proceedings of International Conference on Computing and Communication Networks (pp. 339-353). Springer, Singapore.

--Yang, L., Zhong, C., Yang, Q., Zou, W., & Fathalla, A. (2020). Task offloading for directed acyclic graph applications based on edge computing in Industrial Internet. Information Sciences540, 51-68.

and many more. The differences should be well compared. The bandwidth constraint, storage, computation constraints, delay constraints have been well addressed by some works, e.g.,

--Jin, Z., Zhang, C., Zhao, G., Jin, Y., & Zhang, L. (2021). A context-aware task offloading scheme in collaborative vehicular edge computing systems. KSII Transactions on Internet and Information Systems (TIIS)15(2), 383-403.

--Liu, H., Cao, L., Pei, T., Deng, Q., & Zhu, J. (2019). A fast algorithm for energy-saving offloading with reliability and latency requirements in multi-access edge computing. IEEE Access8, 151-161.

--Shu, C., Zhao, Z., Han, Y., Min, G., & Duan, H. (2019). Multi-user offloading for edge computing networks: A dependency-aware and latency-optimal approach. IEEE Internet of Things Journal7(3), 1678-1689.

--Shu, C., Zhao, Z., Han, Y., & Min, G. (2019, June). Dependency-aware and latency-optimal computation offloading for multi-user edge computing networks. In 2019 16th Annual IEEE International Conference on Sensing, Communication, and Networking (SECON) (pp. 1-9). IEEE.

Similar algorithms can be employed to solve a simplified and simple version of this work. According to the added texts on page 6, Lines 224~227, the methodologies are similar. 

(2) On page 7, in Lines 232~233, the non-convex problem can be directly solved using ADMM, and similar methodologies in the above related works. The reason why designing such genetic algorithm should be well explained.

(3) The experiments are not convincing, lacking detailed parameter setting demonstration, especially for reference [13] and [14] as benchmark algorithms. Because those algorithms are parameter dependent.

Author Response

Dear reviewer,

Thank you very much for providing more valuable references. We have carefully read all the literature you provided and find that most of them aim to solve the task offloading problem, which is different from our work. More specifically, there are three noticeable differences between task offloading and online BranchyNet partitioning proposed in our work:

(1) In problem definition, the task offloading problem emphasizes offloading all or part of computing tasks from an edge device to the associated edge server. And the objective of our work is to make full use of the free resources of edge devices to complete a complex computing task deployed in a given edge environment.

(2) In the system model, all edge devices are independent of each other in the task offloading problem. After all or part of the tasks is offloaded to the edge server, an edge device exchanges with the associated edge server in a one-to-one mode. However, our work aims to distribute an intelligent application to several edge devices with a certain computing capacity through partitioning the BranchyNet. In this way, the selected edge devices will process application requests cooperated.

(3) In optimization objectives, task offloading problem usually sets optimization objectives to minimize the energy consumption or communication delay of all devices in the network. Our work is to optimize the total execution time generated by distributed execution of intelligent applications and the average power consumption on each edge device.

We have made a supplementary description from lines 75 to 83 to show the above differences.

I'm sorry that there was a mistake in the last revision process. The problem solved in this paper is not a non-convex problem but an NP problem as we have already mentioned in line 89. We have corrected this problem and modified the corresponding description from line 237 to 239.

In the comparative experiments, we set up corresponding simulated distributed environments and have given each edge device's processing capacity in GFLOPS and I/O bandwidth settings. When implementing the Neurosurgeon DNN segmentation algorithm, the network structure and performance settings are the same as in our paper. We assume that the current data center load level is 0. In implementing the DAG-based approach, we calculated the local processing time, edge processing time, and output transmission time of each DNN layer based on PALEO with the same setting as in our algorithm. Besides, we constructed the corresponding SDAG according to the method in the original paper. We added relevant descriptions from line 380 to line 386.

The file named "diff.tex" marks up all my revision details. Thank you again for your valuable comments.